# Change in Diet Quality over 12 Years in the 1946–1951 Cohort of the Australian Longitudinal Study on Women’s Health

**DOI:** 10.3390/nu12010147

**Published:** 2020-01-04

**Authors:** Jennifer N. Baldwin, Peta M. Forder, Rebecca L. Haslam, Alexis J. Hure, Deborah J. Loxton, Amanda J. Patterson, Clare E. Collins

**Affiliations:** 1School of Health Sciences, Faculty of Health and Medicine, University of Newcastle, Callaghan, NSW 2308, Australia; jennifer.baldwin@newcastle.edu.au (J.N.B.); rebecca.williams@newcastle.edu.au (R.L.H.); amanda.patterson@newcastle.edu.au (A.J.P.); 2Priority Research Centre for Physical Activity and Nutrition, University of Newcastle, Callaghan, NSW 2308, Australia; 3Research Centre for Generational Health & Ageing, Faculty of Health and Medicine, University of Newcastle, Callaghan, NSW 2308, Australia; peta.forder@newcastle.edu.au (P.M.F.); alexis.hure@newcastle.edu.au (A.J.H.); deborah.loxton@newcastle.edu.au (D.J.L.); 4School of Medicine and Public Health, University of Newcastle, Callaghan, NSW 2308, Australia; 5Hunter Medical Research Institute, New Lambton Heights, NSW 2305, Australia

**Keywords:** diet quality, Australian dietary intake, adults, women’s health

## Abstract

Understanding patterns of dietary change over time can provide important information regarding population nutrition behaviours. The aims were to investigate change in diet quality over 12 years in a nationally representative sample of women born in 1946–1951 and to identify characteristics of women whose diet quality changed over time. The Australian Recommended Food Score (ARFS) was measured in 2001 (*n* = 10,629, mean age 52.1 years) and 2013 (*n* = 9115; *n* = 8161 for both time points) for the mid-aged cohort from the Australian Longitudinal Study on Women’s Health. Participants were categorised by tertiles of baseline diet quality and also classified as ‘diet quality worsened’ (ARFS decrease ≤ −4 points, *n* = 2361), ‘remained stable’ (−3 ≤ change in ARFS ≤ 3 points, *n* = 3077) or ‘improved’ (ARFS increase ≥ 4 points, *n* = 2723). On average, ARFS total and subscale scores remained relatively stable over time (mean [SD] change 0.3 [7.6] points) with some regression to the mean. Women whose diet quality worsened were more likely to be highly physically active at baseline compared with women whose diet quality improved (*p* < 0.001). Among women with poor diet quality initially (lowest baseline ARFS tertile, *n* = 2451, mean [SD] baseline ARFS 22.8 [4.5] points), almost half (47%, *n* = 1148) had not improved after 12 years, with women less likely to be in the healthy weight range (41% compared to 44%) and be never smokers (56% versus 62%, *p* < 0.05) compared with those whose diet improved. Diet quality remained relatively stable over 12 years’ follow up among mid-aged women. Almost half of those with poor baseline diet quality remained poor over time, emphasizing the need to target high-risk groups for nutrition interventions.

## 1. Introduction

Diet quality is an important predictor of all-cause morbidity and mortality [1,2,3]. Diet quality indices such as the Australian Recommended Food Score [4] provide a measure of overall diet variety, nutritional quality, and/or alignment with national dietary guidelines. While higher diet quality is associated with lower all-cause and cardiovascular disease- and cancer-specific morbidity and mortality [1,2,3], as well as lower health care costs [5], an assumption in cohort studies can be that diet quality remains stable over time. 

Some recent studies have investigated the relationship between change in diet quality and health outcomes. Improved diet quality over time, measured using a range of diet quality indices, has been associated with reduced risk of cardiovascular disease [6] as well as total and cause-specific mortality [7,8]. In one study, a 20 percentile points increase in diet quality scores (assessed by the Alternate Healthy Eating Index, Alternate Mediterranean Diet score and the Dietary Approaches to Stop Hypertension score) was associated with an 8–17% reduction in total mortality and a 7–15% reduction in cardiovascular disease-related mortality [8].

Understanding patterns of dietary change over time can provide important information regarding nutrition behaviours and population groups who may require targeted support to improve intakes. Furthermore, identifying high-risk population groups whose diet quality remains poor or deteriorates further over time may assist policy makers in establishing nutrition screening programs and targeted interventions. Therefore, the aim of the current study was to investigate change in diet quality over 12 years in a nationally representative sample of mid-aged Australian women and to examine sociodemographic and health characteristics of women whose diet quality changed over time.

## 2. Materials and Methods 

### 2.1. Australian Longitudinal Study on Women’s Health

This research uses data from the Australian Longitudinal Study on Women’s Health (ALSWH), the methods for which have been described in detail elsewhere and are available at www.alswh.org [9,10,11]. Briefly, over 40,000 Australian women in three age groups (1973–1978 cohort, 1946–1951 cohort and 1921–1926 cohort) were randomly selected from the Medicare database to take part in the baseline survey in 1996. Women living in rural and remote areas were intentionally oversampled to allow sufficient statistical power to analyse data by area of residence. A fourth cohort (born 1989–1995) was added in 2013. Ethical approvals were granted by the University of Newcastle (h−076–0795) and the University of Queensland (200400224). 

### 2.2. Participants: The 1946–1951 Cohort

The 1946–1951 cohort comprised 13,714 women aged 45–50 years old at Survey 1 (1996). For this study, data from Survey 3 (2001) (*n* = 11,228, then aged 50–55 years) and Survey 7 (2013) (*n* = 9151, then aged 62–67 years) were used. The response rates for Survey 3 and Survey 7 were 85% and 81% respectively, excluding women who had died or withdrawn from survey participation since Survey 1 [12].

### 2.3. Sociodemographic Characteristics

The Socioeconomic Indexes for Areas (SEIFA) Index of Relative Socioeconomic Disadvantage was used to rank geographic areas by relative socio-economic disadvantage, where a low score indicates greater disadvantage [13]. The Accessibility Remoteness Index of Australia (ARIA) provided a measure of remoteness in terms of access to service centres, and for this study categories were collapsed into ‘major cities’, ‘inner/outer regional’ or ‘remote/very remote’. Self-reported ability to manage on current income was assessed by a single question and categorised as ‘easy’, ‘not too bad’ or ‘difficult/impossible’.

### 2.4. Health Characteristics

BMI was calculated using self-reported height and weight data [14], and categorised according to World Health Organization recommendations: underweight (<18.5 kg/m^2^), healthy weight (18.5–24.99 kg/m^2^), overweight (25–29.99 kg/m^2^) or obese (≥30 kg/m^2^) [15]. Self-reported smoking status was categorised as ‘never smoked’, ‘ex-smoker’ or ‘current smoker’. Self –reported physical activity level was categorised as ‘sedentary’, ‘low’, ‘moderate’, or ‘high’ based on responses to two survey questions asking about frequency of vigorous and less vigorous exercise. Self-reported general health was collected using a single item question and categorised as ‘excellent/very good’, ‘good’, ‘fair’ or ‘poor’.

### 2.5. Assessment of Dietary Intake

The Dietary Questionnaire for Epidemiological Studies (DQES) Version 2, a validated, semi-quantitative food frequency questionnaire (FFQ), was used to assess dietary intake [16,17]. The DQES uses a 10 point frequency option to ask participants to report their usual consumption of 74 foods and beverages over the past 12 months. Serving sizes are adjusted using portion size photographs. Respondents are asked further questions about the total number of daily serves of fruit, vegetables, bread, dairy products, eggs, fat spreads and sugars. 

### 2.6. Australian Recommended Food Score

The Australian Recommended Food Score (ARFS) was derived using responses to DQES items. To calculate the ARFS, DQES items that were consumed less than once per week scored zero, and those that were consumed once a week, or more, scored one [4]. For the additional questions on type and amount of core foods, a point was added for each of the following responses; at least two fruit serves per day, at least four vegetable serves per day, using reduced fat or skimmed milk, using soy milk, consuming at least 500 mL of milk per day, using high fibre, wholemeal, rye or multigrain breads, having at least four slices of bread per day, using polyunsaturated or monounsaturated spreads or no fat spread, having one or two eggs per week, using ricotta or cottage cheese, using low fat cheese. For alcohol consumption, one point was added for moderate frequency (≤ 4 days per week) and a second point for moderate quantity (1–2 standard drinks, when alcohol was consumed). These additional points were consistent with national dietary intake recommendations [18]. ARFS subscale scores for vegetables (maximum score of 22), fruit (maximum 14), grains (maximum 14), dairy (maximum 7), meat (maximum 5), vegetarian alternatives (e.g., nuts/beans/soy/egg) (maximum 7), fish (maximum 2), fats (maximum 1) and alcohol (maximum 2) were summed to calculate a total ARFS from 0–74, where a higher score represents a higher quality diet. Data from the DQES included at Survey 3 (2001) and Survey 7 (2013) were used to calculate ARFS scores in the current analysis. 

Using data from 93,252 Australians who completed the Healthy Eating Quiz (an online diet-quality self-assessment tool based on the ARFS), among those who completed the survey on two occasions (1.1%, *n* = 1044) individuals in the poorest diet quality group (‘Needs work’, score < 33 points out of maximum 73, *n* = 303) were observed to have a mean increase in their diet quality score of 3.2 (±7.4 points), while those in the ‘Outstanding’ group (score ≥ 47 points, *n* = 97) had a mean decrease in their score of −3.5 (± 7.1 points) [19]. This result informed the a priori decision regarding cut points for defining change, where an increase or decrease of 4 points or more in ARFS was deemed a clinically substantive change in diet quality. For the current analysis, change in ARFS total score was used to categorise participants into three groups: ‘diet quality worsened’ (change in ARFS total score ≤−4 points), ‘diet quality stable’ (−3 ≤ change in ARFS total score ≤3 points) or ‘diet quality improved’ (change in ARFS ≥4 points).

### 2.7. Statistical Analysis

Statistical analyses were conducted using SPSS, version 25.0 (IBM Corp., Armonk, NY, USA). Tests of normality for ARFS data were undertaken by inspecting histograms. Differences in characteristics between groups were investigated using one-way ANOVA or independent samples *t*-tests for continuous variables and using Pearson’s chi-square for categorical variables.

## 3. Results

Sociodemographic, health and diet quality characteristics of women in the 1946–1951 ALSWH cohort at Survey 3 and Survey 7 are summarised in Table 1. On average, at Survey 3, women were aged 52.1 [standard deviation, SD 1.5] years and approximately one-third (34.1%, *n* = 3807) lived in major cities. 

### 3.1. Australian Recommended Food Score

Valid ARFS data were obtained for *n* = 10,629 women at Survey 3 (missing for 1751 women), for *n* = 9115 (missing for 3265 women) women at Survey 7, and for *n* = 8161 at both Surveys 3 and 7. Women who did not have valid ARFS data at one or both time points were more likely to find it difficult/impossible to manage on their current income, live in outer regional Australia and be sedentary, and were less likely to be a non-smoker or be in the healthy BMI category, compared with women with ARFS data at both time points (*p* < 0.01). Of the women lost to follow up at Survey 7, ARFS total scores at Survey 3 were significantly lower for women who died (mean ARFS 31.1 [SD 8.8] points, *n* = 378) and for women who formally withdrew from the study (29.3 [8.0] points, *n* = 786) compared with Survey 7 respondents (33.1 [8.6] points), *p* < 0.05. There was no difference in Survey 3 ARFS total scores for women who became too frail to complete further surveys, (31.1 [8.8] points, *n* = 48), *p* > 0.05.

The mean (SD) ARFS total score was 32.6 (8.8) points at Survey 3 and 33.1 (8.6) points at Survey 7 (Table 1). Overall, ARFS total and subscale scores remained relatively stable over the 12 year study period (Figure 1). The mean (SD) change in ARFS total score was 0.3 (7.6) points, while the mean change in subscale scores ranged from 0.2 (1.2) points for vegetarian alternatives and −0.3 (2.0) points for grains. Differences between scores at the two time points were statistically significant for the total ARFS and all subscales (*p* < 0.01) except fruits (*p* = 0.31), although these differences were not likely clinically significant. 

Women in the lowest ARFS tertile at Survey 3 had significantly lower ARFS at Survey 7 (mean ARFS 26.8 [SD 7.6] points) compared with women in the upper two tertiles (*p* < 0.05) (Table 2). Women in the upper and lower ARFS tertiles at Survey 3 had the greatest change scores between the two time points, with women in the lower tertile increasing their score by 4.2 [7.6] points, and women in the upper tertile decreasing their score by −3.4 [6.9] points, with likely regression to the mean. Compared with women in the upper two tertiles, women in the lowest ARFS tertile at Survey 3 resided on average in areas with greater socioeconomic disadvantage, and were less likely to find it easy to manage on their current income, be never smokers, be highly physically active or report excellent/very good general health (*p* < 0.05) (Table 2).

### 3.2. Characteristics of Women Whose Diet Quality Changed

Across the sample, diet quality worsened for 2361 women (mean [SD] total ARFS change −8.6 [4.4] points), remained stable for 3077 women (−0.01 [2.0]), and improved among 2723 women (8.4 [2.0] points) (Table 3). Compared with women whose diet quality improved, those whose diet remained stable or worsened had a lower mean baseline ARFS, and were more likely to be highly physically active, *p* < 0.05.

Among women in the lowest ARFS tertile at Survey 3 (*n* = 2451), indicating poor baseline diet quality (mean [SD] ARFS 22.8 [4.5] points), almost half (46.8%, *n* = 1148) had remained poor/worsened at Survey 7 (Table 4). Women whose diet quality did improve (*n* = 1303) increased their ARFS by 9.4 [4.6] points. Women whose diet quality remained poor/worsened were less likely to be in the healthy weight range (41% compared to 44%) and be never smokers (56% compared to 62%), *p* < 0.05. 

## 4. Discussion

The current analysis used data from a large nationally representative sample of mid-aged women to investigate potential changes in diet quality over a 12 year period. Overall, diet quality remained relatively stable over time, with change in ARFS score of less than one point across the population sample. This contrasts to a previous cohort study of US adults reporting a steady increase in diet quality measured using the Alternate Healthy Eating Index between 1999–2010 (mean increase 6.9 points on a 110 point scale, linear trend *p* < 0.01), although in that population sample overall diet quality remained relatively low [20]. That study also observed that increases in diet quality were greater among adults with a lower (healthy range) BMI, in concordance with our finding that women who increased their diet quality score from an initially poor diet score were more likely to be in the healthy weight range [20]. However, the association between higher socioeconomic status and greater increases in diet quality reported by that study was not observed in our data. Further comparisons with population-based cohort studies investigating the relationship between change in diet quality and health outcomes is difficult as change in diet quality data has not been reported for the total study samples [6,8]. Among adults with diabetes mellitus, a modest overall improvement in diet quality was observed over a 16 year period measured using the Healthy Eating Index (mean-adjusted increase of three points on a 100 point scale, *p* for trend =0.003) [21].

In the current study, women whose diet quality worsened were more likely to be highly physically active. This is consistent with findings from previous cohort studies reporting slightly lower baseline physical activity among adults with the largest increase in diet quality (measured using the Alternate Healthy Eating Index) [8], although in another study there was no difference in baseline physical activity across quintiles of change using a plant-based diet quality score [7]. Notably, both of these previous studies found that those with the greatest increases in diet quality demonstrated the greatest increases in physical activity over time [7,8]. 

We found that almost half of women with initially poor diets had not improved their diet quality after 12 years. Conversely, those whose diet improved reported an increase in their diet quality score by an average of 8.8 points, resulting in a mean score greater than the sample mean at Survey 7. Contextually, intervention studies reported changes in diet quality scores among people at risk of type 2 diabetes mellitus of +4.3 points using the ARFS (*p* < 0.01 for between group differences) [22], and +4.6 points using the Alternative Healthy Eating Index (*p* < 0.01 for within group differences) [23]. The increase in diet quality in the current study can be at least partly explained by some women who had poor diets at baseline becoming lost to follow up, as baseline ARFS total scores were lower for women who died, withdrew from the study or became frail, compared with women who completed Survey 7. Among people with initially poor diet quality, increases in diet quality scores are associated with a lower risk of death [8]. 

The improvement in diet quality among half of women with initially poor diets reinforces that a significant proportion of the population can make positive behaviour change. Although the reasons for improving/not improving diet quality in this sample remain unknown, this highlights the importance of interventions to improve diet quality in those most at risk. Women whose diet quality did not improve from an initially poor diet were less likely to have ‘never smoked’ or be in the healthy weight range. This is consistent with evidence from the above cohort studies demonstrating a greater proportion of never smokers among adults in the third (representing little change) and fifth (representing the largest increase) quintiles of change in the Alternate Healthy Eating Index compared with those in the first quintile (representing the largest decrease) [8]. This evidence taken together suggests that current and former smokers could potentially benefit most from tailored interventions to improve diet quality, although further investigation into reasons why women’s diet quality remained poor, including underlying health conditions or social circumstances, is warranted.

There are several important limitations to this study. First, although ARFS data were derived from a validated food frequency questionnaire used in the ALSWH, collection of dietary intake via self-report is subject to bias [24]. We previously identified the sub-group least likely to under- or over-report in the mid-aged cohort [4] using the methods of Black [25]. This was based on a ratio of energy intake (EI) to basal metabolic rate, using a mean physical activity level (PAL) of 1.55 for this group, with an FFQ energy intake of 1.27–2.1 times basal metabolic rate (BMR) considered least likely to have mis-reported. After exclusion of FFQ data where the EI/BMR was outside this range the sample was reduced to 2357 surveys (21.1% of the entire sample). Of note is that this limitation has been shown to substantially underestimate relative risks when evaluating associations between diet and health outcomes [26]. Similar results were identified for this sub-group compared with the entire sample for ARFS scores, except that marital status, number of general medical practitioner visits and self-rated health were not significantly associated with quintiles of ARFS [4]. Of note, there was little difference in energy intake across quintiles of ARFS, with the ARFS range similar to the full sample. In addition, results for this ‘valid’ sub-group demonstrated that energy adjustment did not alter findings when reporting results on the relationship between diet quality and health, although the 95% CIs were wider.

Second, as only data for the 1946–1951 cohort of the ALSWH were used, the generalizability of these findings to young or older women, and to men, is limited. Third, the cut-off scores used to develop categories of change in diet quality were based on a sample of Australians that, although large, was not nationally representative [19], and as such the cut-off scores used may not accurately capture clinically important changes in diet quality. Finally, it is likely that the ARFS changes observed can be partly explained by regression to the mean, particularly for the upper/lower extreme baseline scores [27], and this should be taken into account when interpreting results.

## 5. Conclusions

Diet quality remains relatively stable among mid-aged women over time. Almost half of women with poor diets failed to improve their diet quality over time, indicating continued misalignment with Australian Dietary Guidelines. This emphasizes the importance, and potential opportunity, to target population health interventions to improve diet quality for groups most at risk. An understanding of why women’s diet quality remained poor is also needed.

## Figures and Tables

**Figure 1 nutrients-12-00147-f001:**
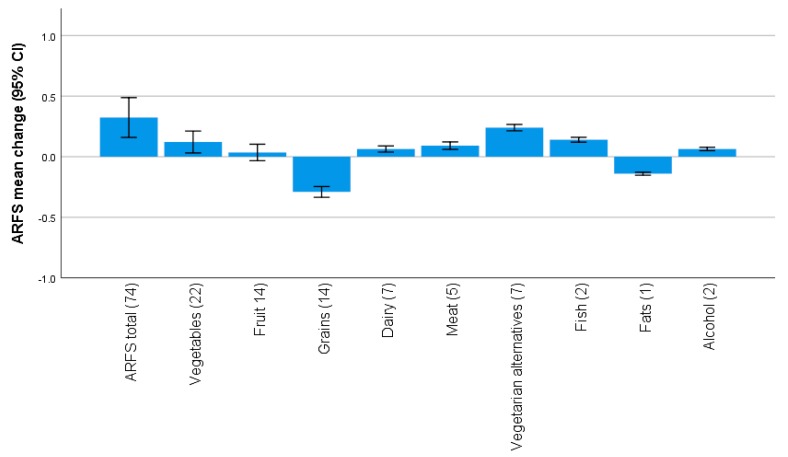
Mean change (95% CI) in Australian Recommended Food Score (ARFS) total and subscale scores (maximum score indicated in brackets) from 2001 to 2013 for the 1946–1951 ALSWH cohort.

**Table 1 nutrients-12-00147-t001:** Characteristics of women in the 1946–1951 cohort of the Australian Longitudinal Study on Women’s Health (ALSWH) at Survey 3 (2001) and Survey 7 (2013).

	Survey 3 (2001) *n* = 11,226	Survey 7 (2013) *n* = 9151
**Sociodemographic Characteristics, % (*n*)**		
Age (years), Mean (SD)	52.1 (1.5)	64.3 (1.5)
SEIFA Index of Disadvantage, Mean (SD)	995.5 (57.5)	998.2 (85.5)
ARIA Classification		
Major Cities	34.1 (3807)	38.3 (3508)
Inner/Outer Regional	61.5 (6861)	58.5 (5354)
Remote/Very Remote	4.4 (485)	2.8 (261)
Ability to Manage on Current Income		
Difficult/Impossible	38.6 (4277)	32.2 (2903)
Not Too Bad	43.5 (4820)	47.9 (4322)
Easy	18.0 (1993)	20.0 (1801)
**Health characteristics, % (*n*)**		
BMI (kg/m^2^), Mean (SD)	26.9 (5.5)	27.8 (5.7)
BMI Classification		
Underweight <18.5 kg/m^2^	1.4 (147)	1.2 (103)
Healthy Weight 18.5–24.99 kg/m^2^	42.5 (4457)	35.0 (3083)
Overweight, 25–29.99 kg/m^2^	32.3 (3388)	33.7 (2971)
Obese, ≥30 kg/m^2^	23.7 (2483)	30.2 (2660)
Smoking Status		
Never Smoked	61.3 (6852)	63.0 (5716)
Ex-Smoker	24.2 (2709)	30.3 (2752)
Current Smoker	14.4 (1618)	6.7 (604)
Physical Activity Level		
Sedentary	18.1 (1936)	17.4 (1530)
Low	37.1 (3968)	24.2 (2127)
Moderate	20.3 (2168)	21.1 (1858)
High	24.5 (2622)	37.3 (3277)
General Health		
Poor	1.6 (175)	1.6 (148)
Fair	12.4 (1380)	12.0 (1099)
Good	39.0 (4347)	39.5 (3605)
Excellent/Very good	47.1 (5250)	46.9 (4279)
**Diet Quality, ARFS Mean Score (SD)**	***n* = 10,629**	***n* = 9115**
Total Score /74	32.6 (8.8)	33.1 (8.6)
Vegetables /22	13.6 (4.4)	13.8 (4.5)
Fruit /14	5.6 (3.2)	5.7 (3.0)
Dairy /7	2.1 (1.0)	2.2 (1.0)
Grains /14	4.0 (1.8)	3.8 (1.8)
Meat /5	2.4 (1.3)	2.5 (1.3)
Vegetarian Alternatives /7	2.0 (1.1)	2.3 (1.1)
Fish /2	0.9 (0.8)	1.0 (0.8)
Fats /1	0.7 (0.5)	0.5 (0.5)
Alcohol /2	1.4 (0.6)	1.5 (0.6)

Abbreviations: ARFS, Australian Recommended Food Score; ARIA, Accessibility Remoteness Index of Australia; BMI, Body Mass Index; SEIFA, Socioeconomic Index for Areas; SD, standard deviation.

**Table 2 nutrients-12-00147-t002:** Characteristics of women in the 1946–1951 ALSWH cohort by tertile of baseline diet quality.

	2001 ARFS Tertile		
	Tertile 1 (Lowest) *n* = 3357	Tertile 2 *n* = 3697	Tertile 3 (Highest) *n* = 3575
2001 ARFS Total, Mean (SD)	22.5 (4.7)	32.5 (2.2)	42.2 (4.3)
2013 ARFS Total, Mean (SD)	26.8 (7.6)	33.3 (6.9) *	38.7 (7.0) *
Change in ARFS Total 2001 to 2013, Mean (SD)	4.2 (7.1)	0.7 (6.8) *	−3.4 (6.9) *
**Sociodemographic Characteristics (Survey 3 2001), % (*n*)**			
SEIFA Index of Disadvantage, Mean (SD)	990.6 (56.0)	996.9 (58.5) *	999.5 (57.8) *
Living in Major Cities	33.0 (1100)	34.9 (1282)	34.4 (1223)
Managing on Current Income Easy/Not Too Bad	56.2 (1863)	62.8 (2291) *	65.7 (2320) *
**Health Characteristics (Survey 3 2001), % (*n*)**			
BMI (kg/m^2^), Mean (SD)	27.0 (5.8)	26.8 (5.4)	26.8 (5.2)
Healthy Weight Range for BMI	41.9 (1297)	42.8 (1491)	42.8 (1491)
Never Smoker	56.9 (1903)	61.4 (2253) *	64.7 (2299) *
High Physical Activity	19.3 (614)	23.4 (826) *	30.9 (1066)
Excellent/Very Good General Health	41.4 (1380)	47.5 (1739) *	53.1 (1883) *

Abbreviations: ARFS, Australian Recommended Food Score; BMI, body mass index; SD, standard deviation; SEIFA, Socioeconomic Index for Areas. * Different from Tertile 1, *p* < 0.05.

**Table 3 nutrients-12-00147-t003:** Characteristics of women in the 1946–1951 ALSWH cohort by category of change in diet quality.

	Diet Quality Got Worse ^a^ (*n* = 2361)	Diet Stayed the Same ^b^ (*n* = 3077)	Diet Quality Improved ^c^ (*n* = 2723)
Change in ARFS Total 2001 to 2013, Mean (SD)	−8.6 (4.4)	−0.01 (2.0)	8.4 (2.0)
2001 ARFS Total, Mean (SD)	37.4 (8.0) **	33.2 (8.0) **	28.6 (7.7)
2013 ARFS Total, Mean (SD)	28.9 (8.4) **	33.2 (8.0) **	37.0 (7.5)
**Sociodemographic Characteristics (2001), % (*n*)**			
SEIFA Index of Disadvantage, Mean (SD)	997.5 (56.8)	997.9 (58.4)	995.3 (58.1)
Living in Major Cities	33.7 (793)	33.9 (1041)	35.3 (957)
Managing on Current Income Easy/Not Too Bad	63.4 (1485)	65.6 (1997)	62.8 (1698)
**Health Characteristics (Survey 3, 2001), % (*n*)**			
BMI (kg/m^2^), Mean (SD)	26.9 (5.4)	26.5 (5.2)	26.7 (5.4)
Healthy Weight Range for BMI	40.2 (947)	44.2 (1281)	44.2 (1135)
Never Smoker	61.7 (1454)	62.9 (1928)	63.3 (1721)
High Physical Activity	29.7 (676) **	24.3 (727) *	22.8 (593)
Excellent/Very Good General Health	50.9 (1192)	50.8 (1556)	50.1 (1356)

Abbreviations: ARFS, Australian Recommended Food Score; BMI, body mass index; SD, standard deviation; SEIFA, Socioeconomic Index for Areas. ^a^ Change in ARFS total score ≤−4 points. ^b^ −3 ≥ Change in ARFS total score ≤3 points. ^c^ Change in ARFS total score ≥4 points. * Different from ‘Diet quality improved’, *p* < 0.05. ** Different from ‘Diet quality improved’, *p* < 0.01.

**Table 4 nutrients-12-00147-t004:** Characteristics of women in the lowest baseline ARFS tertile (*n* = 2451) by category of change in diet quality.

	Change in ARFS for Women in the Lowest Baseline ARFS Tertile (*n* = 2451)	
	Diet Quality Remained Poor/Worsened ^a^ (*n* = 1148)	Diet Quality Improved ^b^ (*n* = 1303)
2001 ARFS Total, Mean (SD)	23.4 (4.1)	22.0 (4.8)
2013 ARFS Total, Mean (SD)	21.6 (5.6)	31.5 (5.9)
Change in ARFS Total, Mean (SD)	−1.8 (4.1)	9.4 (4.6)
**Sociodemographic Characteristics (Survey 3 2001) % (*n*)**		
SEIFA Index of Disadvantage, Mean (SD)	993.9 (57.5)	991.9 (56.0)
Living in Major Cities	35.1 (401)	33.2 (430)
Managing on Current Income Easy/Not Too Bad	58.3 (660)	58.3 (754)
**Health Characteristics (Survey 3 2001) % (*n*)**		
BMI (kg/m^2^), Mean (SD)	26.9 (5.8)	26.9 (5.7)
Healthy Weight Range for BMI	41.1 (434) *	44.2 (540)
Never Smoker	55.6 (637) *	61.5 (800)
High Physical Activity	19.7 (218)	19.3 (241)
Excellent/Very Good General Health	42.9 (489)	46.8 (607)

Abbreviations: ARFS, Australian Recommended Food Score; BMI, body mass index; SD, standard deviation; SEIFA, Socioeconomic Index for Areas. ^a^ Change in ARFS total score ≤3 points ^b^ Change in ARFS total score ≥4 points * Different from ‘Diet quality improved’, *p* < 0.05

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
