# Peer review of "Change in Diet Quality over 12 Years in the 1946–1951 Cohort of the Australian Longitudinal Study on Women’s Health"

_nutrients, 2020, doi:10.3390/nu12010147_

Round 1
Reviewer 1 Report
General comments:
In their study, the authors employed a method (FFQs) that has been demonstrated to be invalid, yet the authors failed to cite or even acknowledge the large body of contrary work.
This type of behavior is not acceptable because it is extremely misleading for authors to ignore research conducted by dozens of nutrition & obesity researchers and cited thousands of times, inclusive of respected journals such as The Lancet, International Journal of Obesity, Mayo Clinic Proceedings, and the Journal of Clinical Epidemiology.
For a series of papers detailing the debate please see [1-4]:
Archer, E., et al. (2018). "Controversy and Debate: Memory-Based Methods Paper 1: The Fatal Flaws of Food Frequency Questionnaires and other Memory-Based Dietary Assessment Methods." Journal of Clinical Epidemiology 104(December): 113-124. Martin-Calvo, N. and M. A. Martinez-Gonzalez (2018). "Controversy and debate: Memory-Based Dietary Assessment Methods Paper 2." J Clin Epidemiol 104: 125-129. Archer, E., et al. (2018). "Controversy and Debate: Memory-Based Dietary Assessment Methods Paper #3." Journal of Clinical Epidemiology 104(December): 130-135. Martin-Calvo, N. and M. A. Martinez-Gonzalez (2018). "Controversy and debate: Memory-Based Methods Paper 4." J Clin Epidemiol 104: 136-139.For general reviews of the debate see [5, 6]
Archer, E., et al. (2018). "The Failure to Measure Dietary Intake Engendered a Fictional Discourse on Diet-Disease Relations." Frontiers in Nutrition 5(105). Dhurandhar, N. V., et al. (2014). "Energy balance measurement: when something is not better than nothing." Int J Obes (Lond) 39(7): 1109-1113. This latter paper is the work of 50+ investigators and has been cited over 300 times.To be precise, the current debate does not revolve around measurement error or mere bias. The debate stems from logical fallacies, and the non-falsifiability and pseudo-quantification of self-reported (M-BMs) data.
It is unequivocal that there are large and clinically significant qualitative and quantitative differences between what people remember eating and drinking, what they are willing and able to report eating and drinking, and their actual dietary intake.[7-9]
This fact was repeatedly replicated over the past five-decades, and the incongruity between anecdotal reports and objective evidence is explained by intentional and non-intentional factors that distort recall such as confabulation, lying, forgetting, false memories, social desirability, impression management, ‘mindless eating’, and the failure to encode, remember, and recall consumptive behaviors. (Please see [8, 9] for review).
Two questions the authors need to address:
First: Given that people lie, misremember, and are prone to social desirability and impression management, how can the authors demonstrate that any of their self-reported data are accurate? The fact that people lie about their diet, weight, height, physical activity, and smoking is too well-established to continue to be ignored by nutrition epidemiologists.
Second: Given that M-BMs data and biochemical analyses differ systematically for energy (calories) and most nutrients, how can the authors support the pseudo-quantification of dietary self-reports and assume their data are plausible w/ respect to health?
The use objective data (i.e., biochemical analyses) eliminates the issues with M-BMs. Thus, the use of physiologically implausible dietary self-reports is an anachronism from the previous century and the field must acknowledge this and move forward.
A general question: Is 'diet-centrism' a valid perspective from which to view public health? [10, 11] For example, if a population’s diet is unhealthy, then age-adjusted non-communicable chronic diseases should be increasing. Yet in most industrialized nations age-adjusted rates of cancer and CVD have decreased for many decades. Thus, it appears that the effect of current diet on cancer and CVD is either positive or trivial given that in the U.S., populations with the worst dietary patterns (e.g., African Americans) have exhibited the largest gains in health- and life-spans over past four decades despite consuming a diet that is assumed to be inferior (i.e., ‘unhealthy’; e.g., more fast food, less fruits and vegetables, more saturated fat and sugar, etc.).
In summary, the present study offers no data on actual diet. It presents data on reported memories of perceptions of consumption with no attempt to acknowledge or address the actual debate on the validity of M-BMs or how the demonstrated lack of validity affects their conclusions.
References
Martin-Calvo, N. and M.A. Martinez-Gonzalez, Controversy and debate: Memory-Based Methods Paper 4. J Clin Epidemiol, 2018. 104: p. 136-139. Archer, E., M.L. Marlow, and C.J. Lavie, Controversy and Debate: Memory-Based Dietary Assessment Methods Paper #3. Journal of Clinical Epidemiology, 2018. 104(December): p. 130-135. Archer, E., M.L. Marlow, and C.J. Lavie, Controversy and Debate: Memory-Based Methods Paper 1: The Fatal Flaws of Food Frequency Questionnaires and other Memory-Based Dietary Assessment Methods. Journal of Clinical Epidemiology, 2018. 104(December): p. 113-124. Martín-Calvo, N. and M.Á. Martínez-González, Controversy and debate: Memory-Based Dietary Assessment Methods paper 2. Journal of Clinical Epidemiology, 2018. 104: p. 125-129. Archer, E., C.J. Lavie, and J.O. Hill, The Failure to Measure Dietary Intake Engendered a Fictional Discourse on Diet-Disease Relations. Frontiers in Nutrition, 2018. 5(105). Dhurandhar, N.V., et al., Energy balance measurement: when something is not better than nothing. Int J Obes (Lond), 2014. 39(7): p. 1109-1113. Archer, E., G.A. Hand, and S.N. Blair, Validity of U.S. Nutritional Surveillance: National Health and Nutrition Examination Survey caloric energy intake data, 1971-2010. PLoS One, 2013. 8(10): p. e76632. Archer, E., G. Pavela, and C.J. Lavie, The Inadmissibility of What We Eat in America and NHANES Dietary Data in Nutrition and Obesity Research and the Scientific Formulation of National Dietary Guidelines. Mayo Clinic Proceedings, 2015. 90(7): p. 911-26. Archer, E., G. Pavela, and C.J. Lavie, A Discussion of the Refutation of Memory-Based Dietary Assessment Methods (M-BMs): The Rhetorical Defense of Pseudoscientific and Inadmissible Evidence. Mayo Clinic Proceedings, 2015. 90(12): p. 1736-1738. Archer, E. (2018). "In Defense of Sugar: A Critique of Diet-Centrism." Progress in Cardiovascular Diseases 61(1): 10-19. Archer, E. (2018). "The Demonization of ‘Diet’ Is Nothing New." Progress in Cardiovascular Diseases 61(3 (September/October)).
Author Response
Reviewer 1
…In summary, the present study offers no data on actual diet. It presents data on reported memories of perceptions of consumption with no attempt to acknowledge or address the actual debate on the validity of M-BMs or how the demonstrated lack of validity affects their conclusions.
Author response: We acknowledge the concerns raised by Reviewer 1 and with references to current literature, summarised in the above item. We would like to respond in turn:
First, we recognise that “there will always be error in dietary assessments. The challenge is to understand, estimate, and make use of the error structure during analysis” (Beaton GH, Burema J, Ritenbaugh C. Am J Clin Nutr 1997;65(4, Suppl)1100S–7S). In their commentary on concerns regarding self-reported dietary data, Amy Subar and colleagues discuss the types of measurement error – random and systematic – and observe that all measurements, whether self-reported or objective, can be subject to both types of error. Nutrition scientists have developed numerous methods to acknowledge and minimise error in self-reported dietary data (e.g. Neuhouser ML, et al. Am J Epidemiol 2008;167:1247–59.)
Second, in this manuscript we have ranked women according to quintiles of diet quality. This approach is common practice in nutrition research (e.g. Schwingshackl L and Hoffmann G. Journal of the Academy of Nutrition and Dietetics, 2015. 115(5): p. 780-800. e5) as it enables comparison of relative diet quality rather than drawing conclusions about absolute food/nutrient intake.
Third, we have investigated the relationship between self-reported diet quality (ARFS) and plasma carotenoids in a sample of 99 overweight/obese adults (Ashton L, et al. Nutrients 2017;9(8):888). In this study we used three statistical approaches to examine the relationship between ARFS and plasma carotenoids: i) Spearman rank correlations, ii) linear regression models adjusting for total energy intake, total fat intake, BMI and sex, and iii) weighted Kappa statistic to measure agreement. We observed moderate correlations (r >0.20) were found between the ARFS fruit subscale score and α-carotene, β-carotene, and β-cryptoxanthin, and between the combined fruit and vegetable score and plasma concentrations of α-carotene and β-carotene. Results from linear regression showed that the ARFS fruit subscale explained the variation in α-carotene (p < 0.001) and β-carotene (p < 0.05), while the ARFS combined fruit and vegetables score explained the variation in α-carotene (p < 0.001). Significant agreement was found between ARFS, ARFS fruit and ARFS fruit + vegetable subscales.
Finally, in our manuscript we have acknowledged to the reader the inherent limitations of collecting dietary intake data by self-report (Section 4, Lines 280-282).
We thank Reviewer 1 for their detailed response to our manuscript, and hope that our response above is sufficient to provide a balanced consideration of the limitations of our study.
Manuscript change: No manuscript changes made.

Reviewer 2 Report
Abstract
Line 17 – how or whether the diet changes is more than rarely reported. The NHANES in the US for example has extensive data on changes in diet. This statement is not correct or is certainly open to misinterpretation if the authors have another definition in mind what constitutes population level analyses of dietary trends over time.
Introduction
Line 42 - an assumption in cohort studies… In reference to the point above about reporting of dietary changes over time, this statement has issues. Either provide references to support such a statement or revise. It’s well documented and acknowledged that diets have changed – not just in Australia but globally – over time. The more relevant questions are – are those changes good or bad?
Materials and methods
2.6 line 102 – For fruits - does this score differentiate between fresh and processed fruits? For example a fresh pear or a pear sweetened with sugar in a can? For grains – does this score factor in the type of grain – white bread and multigrain bread are both grains.
2.6 line 117 – what % of the total survey population does this represent? An online quiz may have a respondent bias
Discussion
Line 229 – do you mean LESS physically active?
Author Response
Reviewer 2
Abstract, Line 17 – how or whether the diet changes is more than rarely reported. The NHANES in the US for example has extensive data on changes in diet. This statement is not correct or is certainly open to misinterpretation if the authors have another definition in mind what constitutes population level analyses of dietary trends over time.Author response: We have changed the introductory line in the abstract to more accurately reflect the background to this study, taking into account current literature as indeed the NHANES has contributed substantially to our knowledge of dietary behaviours over time.
Manuscript change: Abstract, Lines 17-19, as follows: “Understanding patterns of dietary change over time can provide important information regarding population nutrition behaviours”
Introduction, Line 42 - an assumption in cohort studies… In reference to the point above about reporting of dietary changes over time, this statement has issues. Either provide references to support such a statement or revise. It’s well documented and acknowledged that diets have changed – not just in Australia but globally – over time. The more relevant questions are – are those changes good or bad?
Author response: We acknowledge that population dietary patterns do indeed change over time (e.g. Rehm CD, et al. JAMA 2016;315(23):2542-53). We have made the statement in this regard at Lines 44-45: ‘…an assumption in cohort studies can be that diet quality remains stable over time’ as we are specifically referring to the cohort studies referenced earlier in the sentence that have reported associations of diet quality (measured at one time-point) on health outcomes and healthcare costs over time. We agree that population level diets have changed over recent decades in many countries but feel that such commentary is outside the scope of the Introduction here as we are building the case for examining diet quality over time in this cohort of Australian women.
Manuscript change: No changes made.
Section 2.6 line 102 – For fruits - does this score differentiate between fresh and processed fruits? For example a fresh pear or a pear sweetened with sugar in a can? For grains – does this score factor in the type of grain – white bread and multigrain bread are both grains.
Author response: In the original paper documenting construction of the ARFS (Collins, Young & Hodge 2008), more detailed description of the scoring method is provided. The scoring is modelled on the US recommended Food Score (Kant AK, Thompson FE. Nutrition Research 1997;17:1443-56) and focusses on a variety of healthful foods but does not distinguish canned versus fresh. For fruit, one point was scored for consuming each of the following at least once a week, Canned or frozen Fruit; Oranges or other citrus; Apples; Pears; Bananas; Melons (water, rock, honeydew); Pineapple; Strawberries; Apricots; Peach/nectarines; Mango/pawpaw; Avocado. Fruit or Vegetable Juice. We have added additional information in our manuscript to improve clarity for the reader.
Manuscript change: Section 2.6, Lines 104-115; as follows:-
“For the additional questions on type and amount of core foods, a point was added for each of the following responses; at least two fruit serves per day, at least four vegetable serves per day, using reduced fat or skimmed milk, using soy milk, consuming at least 500mL of milk per day, using high fibre, wholemeal, rye or multigrain breads, having at least four slices of bread per day, using polyunsaturated or monounsaturated spreads or no fat spread, having one or two eggs per week, using ricotta or cottage cheese, using low fat cheese.”
Section 2.6 line 117 – what % of the total survey population does this represent? An online quiz may have a respondent bias.
Author response: Of the total respondents to the quiz, 1.1% completed the survey on two occasions. We agree that this online quiz could have had a respondent bias, and we acknowledge this as a limitation to the study in the final paragraph of the Discussion.
Manuscript change: Section 2.6, Line 128, as follows: “1.1%” added in brackets.
Discussion, Line 229 – do you mean LESS physically active?
Author response: Women whose diet quality worsened were more likely to be highly physically active (29.7% highly physically active) compared to those whose diet quality improved (22.8%) or remained stable (24.3%). This is an interesting finding and somewhat counter-intuitive (one would think worsening diet quality would be associated with lower physical activity), but is consistent with findings of Sotos-Prieto and colleagues that adults with the largest increase in diet quality had slightly lower baseline physical activity. Their study also found that adults with the greatest increase in diet quality also had the greatest increase in physical activity, although we did not investigate change in physical activity over time in our study.
Manuscript change: No changes made.

Reviewer 3 Report
Thank you very much for your important investigation. The study is done excellent. Nevertheless I suggest some supplementation/changes
It is important to have a look at women’s health and it seems very practical to use data from a large study already existing. Here you investigate just women’s health. But men’s health is as important as women’s health. Please give an explanation why you or why the Australian Longitudinal Study just addressed women. Tables: As you mentioned in the text you figured p-values for all comparisons of the different groups in your tables. Please add them in an additional column the tables as well. Materials and Methods: in line 113-124 your are giving results. They belong to the result section. Please move them. Results: Line 135-139 repeats results the table 1. You can erase these sentences.Author Response
Reviewer 3
It is important to have a look at women’s health and it seems very practical to use data from a large study already existing. Here you investigate just women’s health. But men’s health is as important as women’s health. Please give an explanation why you or why the Australian Longitudinal Study just addressed women.
Author response: We agree that men’s health is equally important, and there are several other Australian cohort studies investigating health outcomes for both sexes (e.g. 45 and Up study) and for men only (e.g. CHAMP study). We have used data for the ASLWH which was instigated by the Australian Government to specifically look at the health of Australian women and to address knowledge gaps in women’s health.
Manuscript change: No changes made.
Tables: As you mentioned in the text you figured p-values for all comparisons of the different groups in your tables. Please add them in an additional column the tables as well.
Author response: We feel that we have adequately indicated statistical significance in the Tables using asterisk marks (* for p<0.05 and ** for p<0.01) and would prefer our Tables to remain as is, if you agree. However we have realised an error in the footer for Table 3 and have made changes accordingly.
Manuscript change: Table 3 footer, Lines 206-207, as follows: * Different from Tertile 1 ‘Diet quality improved’, p<0.05; ** Different from ‘Diet quality improved’, p<0.01
Materials and Methods: in line 113-124 you are giving results. They belong to the result section. Please move them.
Author response: We have moved the lines in question from the Methods to the Results section.
Manuscript change: Section 2.6, Lines 120-125 deleted and added to Section 3.1, Lines 156-161.
Results: Line 135-139 repeats results the table 1. You can erase these sentences.
Author response: We have made changes to avoid replication of results in the text and tables.
Manuscript change: Section 3, Lines 145-148. Paragraph to now read as: “Sociodemographic, health and diet quality characteristics of women in the 1946-51 ALSWH cohort at Survey 3 and Survey 7 are summarised in Table 1. On average at Survey 3, women were aged 52.1 [standard deviation, SD 1.5] years and approximately one-third (34.1%, n=3807) of participants lived in major cities.”

Round 2
Reviewer 1 Report
The author's statement "there will always be error in dietary assessments" misses the point of the original critique entirely. If this was an honest error from a lack of due diligence, it can be corrected simply by addressing the points in the original review (also summarized below).
A summary of the previous review:
The measurement errors associated with self-reported data are non-quantifiable and are thus pseudo-scientific (i.e., non-falsifiable). This is not a mere limitation. It is a refutation of the validity of the methods and data employed in the present study. Nevertheless, the authors simply ignored this critique (and fact). To reiterate the main point, empirical refutations are NOT mere limitations and need to be addressed explicitly in the manuscript. I fully understand the author's position, but the field of nutrition cannot continue to ignore contrary evidence and present mere associations from implausible data, while pretending that they are contributing to public health science. For example, the author's statement that the correlation of 0.20 is "moderate" is truly risible; less than 5% of the variance is being 'explained' by the association. What explains the other 95%? If self-reported data are a proxy for actual consumption, then the correlations should be between .7 and .9 as in other medical and scientific fields. A thermometer with a correlation of .5 would lead to misdiagnoses and death. More importantly, absolute values are critical to nutrient deficiencies and health, yet the field of nutrition appears to be content publishing ridiculously low correlations as valid evidence. FFQs and other memory-based dietary assessment methods (M-BMs) cannot assign participants to the proper tertile with respect to most dietary factors (e.g., calories, sodium, potassium, etc.). So why do the authors consider M-BMs data valid? Finally, there is a large body of work cited thousands of times supporting the critiques made in the original review. Importantly, The Lancet, one of the top journals in the world recently published a critique of Walter Willet's work that called self-reported dietary data "meaningless". If the Lancet is willing to acknowledge the lack of validity of M-BMs, why do the authors think they can simply ignore this large and growing body of contrary work?
Author Response
Reviewer 1 Comments
1.The author's statement "there will always be error in dietary assessments" misses the point of the original critique entirely. If this was an honest error from a lack of due diligence, it can be corrected simply by addressing the points in the original review (also summarized below).
A summary of the previous review:
The measurement errors associated with self-reported data are non-quantifiable and are thus pseudo-scientific (i.e., non-falsifiable). This is not a mere limitation. It is a refutation of the validity of the methods and data employed in the present study. Nevertheless, the authors simply ignored this critique (and fact).
Author Response
Self-report food frequency questionnaire (FFQ) data is prone to error, as is all self-reported data including data related to other aspects of health and lifestyle including physical activity and smoking. A detailed review of FFQ methodology is beyond the scope of the current paper but has been detailed elsewhere (Subar et al 2017) and we have now referred the reader to this publication, published in the American Journal of Epidemiology in 2017.
We have previously evaluated the level of energy mis-reporting in the mid-aged cohorts (Collins et al 2007) and identified the group least likely to under- or over-report based on ratio of energy intake (EI) to basal metabolic rate, using a mean physical activity level (PAL) of 1.55 for this group, with an FFQ energy intake of 1.27–2.1 times BMR considered plausible and least likely to have mis-reported their dietary intake, using the method of Black (Black 2000). After exclusion of FFQ data where the EI/BMR was outside this range the sample was reduced to 2357 surveys (21.1% of the entire sample). Similar results were found for this sub-group compared with the entire group of women with ARFS scores, except that marital status, number of general medical practitioner visits and self-rated health were not significantly associated with quintiles of ARFS. In addition, there was little difference in energy intake across quintiles of ARFS, with the ARFS range similar to that in the full sample. ALSWH results for this valid sub-group demonstrated that energy adjustment did not alter findings when reporting results on the relationship between diet quality and health, although the 95% CIs were wider.
Of note is that this limitation has been shown to substantially underestimate relative risks in association between diet and health outcomes and to reduce statistical power for detecting associations.
Manuscript change: We have added the above information to the discussion at Lines 282-295, as follows:-
We previously identified the sub-group least likely to under- or over-report in the mod-aged cohort (Collins et al 2007) using the methods of Black (Black 2000). This was based on a ratio of energy intake (EI) to basal metabolic rate, using a mean physical activity level (PAL) of 1.55 for this group, with an FFQ energy intake of 1.27–2.1 times BMR considered least likely to have mis-reported. After exclusion of FFQ data where the EI/BMR was outside this range the sample was reduced to 2357 surveys (21.1% of the entire sample). Of note is that this limitation has been shown to substantially underestimate relative risks when evaluating associations between diet and health outcomes (Freedman 2011).
Similar results were identified for this sub-group compared with the entire sample for ARFS scores, except that marital status, number of general medical practitioner visits and self-rated health were not significantly associated with quintiles of ARFS (Collins et al 2007). Of note, there was little difference in energy intake across quintiles of ARFS, with the ARFS range similar to the full sample. In addition, results for this ‘valid’ sub-group demonstrated that energy adjustment did not alter findings when reporting results on the relationship between diet quality and health, although the 95% CIs were wider.
References
Collins CE, Young AF, Hodge A. Diet Quality Is Associated With Higher Nutrient Intake and Self‑Rated Health In Mid‑Aged Women. Journal of the American College of Nutrition, 2008 Feb;27(1):146-57.
Black AE: Critical evaluation of energy intake using the Goldberg cut-off for energy intake: basal metabolic rate. A practical guide to its calculation, use and limitations. Int J Obes Relat Metab Disord 24:1119–1130, 2000.
Freedman LS, Schatzkin A, Midthune D, et al. Dealing with dietary measurement error in nutritional cohort studies. J Natl Cancer Inst. 2011;103(14):1086–1092.
To reiterate the main point, empirical refutations are NOT mere limitations and need to be addressed explicitly in the manuscript. I fully understand the author's position, but the field of nutrition cannot continue to ignore contrary evidence and present mere associations from implausible data, while pretending that they are contributing to public health science. For example, the author's statement that the correlation of 0.20 is "moderate" is truly risible; less than 5% of the variance is being 'explained' by the association. What explains the other 95%? If self-reported data are a proxy for actual consumption, then the correlations should be between .7 and .9 as in other medical and scientific fields. A thermometer with a correlation of .5 would lead to misdiagnoses and death. More importantly, absolute values are critical to nutrient deficiencies and health, yet the field of nutrition appears to be content publishing ridiculously low correlations as valid evidence. FFQs and other memory-based dietary assessment methods (M-BMs) cannot assign participants to the proper tertile with respect to most dietary factors (e.g., calories, sodium, potassium, etc.). So why do the authors consider M-BMs data valid?
Author Response
The question as to the level of correlation required to ‘validate’ an FFQ for specific has been raised elsewhere, (Subar 2017) and critiqued by others in details (Byers 2001) and is beyond the scope of the current paper.
Limitations impacting on the correlation between dietary intake assessed by self-report (recalled) methods and prospective methods are affected by factors including differences in nutrient database values, which are based on representative food samples, and actual composition of specific foods consumed at any point in time. Nutrient composition of individual foods is also impacted by issues such as where food is grown/produced, soil nutrient content, food preparation techniques and cooking methods.
Data from the original validation of the FFQ compared to weighed food records demonstrated correlation coefficients ranged from 0.28 for vitamin A to 0.78 for carbohydrate (Hodge 2001), in the range of that reported for other instruments indicating it is just as good or bad as other instruments. While noting that weight food records also have limitations. At individual level, discrepancies are not particularly meaningful for population data and we would be more concerned if there was evidence of systematic bias. As highlighted above, limitations in use of FFQs when compared to biomarker data have been show to attenuate diet-disease relationships towards the null (Subar 2017).
Reference
Subar AF, Kushi LH, Lerman JL, Freedman LS. Invited Commentary: The Contribution to the Field of Nutritional Epidemiology of the Landmark 1985 Publication by Willett et al. Am J Epidemiol. 2017 Jun 1;185(11):1124-1129.
Byers T. Food frequency dietary assessment: how bad is good enough? Am J Epidemiol. 2001 Dec 15;154(12):1087-8.
Hodge A, Patterson AJ, Brown WJ, Ireland P, Giles G. The Anti Cancer Council of Victoria FFQ: relative validity of nutrient intakes compared with weighed food records in young to middle-aged women in a study of iron supplementation. Aust N Z J Public Health. 2000 Dec;24(6):576-83.
Finally, there is a large body of work cited thousands of times supporting the critiques made in the original review. Importantly, The Lancet, one of the top journals in the world recently published a critique of Walter Willet's work that called self-reported dietary data "meaningless". If the Lancet is willing to acknowledge the lack of validity of M-BMs, why do the authors think they can simply ignore this large and growing body of contrary work?
Author response
Interestingly The American Journal of Epidemiology (Subar 2017), a top journal for epidemiology, published a commentary on the important impact of Willet’s development of FFQs on the field of nutritionally epidemiology. Subar’s 2015 response to current criticism has been cited hundreds of times since 2015. Together these highlight that there are opposing opinions in this field, as there are in many other fields.
References
Subar AF, Kushi LH, Lerman JL, Freedman LS. Invited Commentary: The Contribution to the Field of Nutritional Epidemiology of the Landmark 1985 Publication by Willett et al. Am J Epidemiol. 2017 Jun 1;185(11):1124-1129.
Subar AF, Freedman LS, Tooze JA, Kirkpatrick SI, Boushey C, Neuhouser ML, Thompson FE, Potischman N, Guenther PM, Tarasuk V, Reedy J, Krebs-Smith SM. Addressing Current Criticism Regarding the Value of Self-Report Dietary Data. J Nutr. 2015 Dec;145(12):2639-45.
